# A Qualitative Study to Understand the Impact of Caring for Traumatic Injury Survivors

**DOI:** 10.3390/ijerph192316202

**Published:** 2022-12-03

**Authors:** Catherine Hudson, Kate Radford, Jade Kettlewell

**Affiliations:** 1Centre for Rehabilitation and Ageing Research, University of Nottingham, Nottingham NG7 2RD, UK; 2Centre for Academic Primary Care, University of Nottingham, Nottingham NG7 2RD, UK

**Keywords:** traumatic injuries, informal carers, carer burden

## Abstract

Background: Following traumatic injury, an informal carer is often required to support recovery. Understanding the impact of caregiving is important to inform intervention design. Aim: to explore the impact of caring on family and caregiver finances, employment, social life, and psychological wellbeing. Method: Semi-structured interviews conducted with carers of traumatic injury survivors. Interviews were audio recorded, transcribed and thematically analysed, informed by the Roy Adaptation Model (RAM) and International Classification of Functioning, Disability and Health (ICF). Results: Ten participants were interviewed. Key themes included (1) financial impact/employment issues, (2) relationships and support and (3) psychological impact. Most carers did not receive professional support with daily care post-discharge. Carers’ employers responded positively, supporting them even after extensive leave. Carers received inconsistent communication whilst visiting trauma survivors in hospital; carers with healthcare experience were favoured. Navigating and receiving benefits was complex. Some carers found it difficult to accept the trauma survivor’s injury, whilst others focused on achieving goals. Conclusions: Support from professional services is limited outside hospital settings for non-brain injuries. Future interventions and healthcare services should acknowledge the lack of psychological support for carers. Researchers should consider using the ICF/RAM when designing interventions to ensure the full impact on carers is addressed.

## 1. Introduction

Traumatic injuries affect a significant proportion of individuals worldwide. Injuries can be life changing, with many survivors experiencing long-term issues [1,2,3], placing significant burden on families. Trauma survivors often need the support of an informal carer; defined as an unpaid relative/friend who lives with and supports their physical daily living. Approximately one in eight adults (equating to around 6.5 million) in the UK are unpaid carers, with 5 million of these individuals juggling caring responsibilities alongside work [4] (carers UK). Research specific to traumatic injury indicates that the financial costs to carers are unknown [5]. The sudden and serious nature of major traumatic injury and change in life circumstances mean caregivers are often required to take time off work [6], sometimes leading to financial problems. Research suggests that as many as 61% of carers of people with a traumatic brain injury (TBI) leave their jobs [7], although UK-specific figures are unknown.

Caring for someone following injury can impact on the carer’s psychological wellbeing, social life, and family and caregiver finances. Some of these areas are well researched; studies investigating the psychological or emotional consequences of being a caregiver have found that social support contributes significantly to the variance in caregiver depression [8,9]. Fewer studies have investigated carers’ response to support in the immediate aftermath of moderate to severe traumatic injury. In a study of 184 TBI carers, O’Callaghan et al. found carer anxiety to be related to a lack of information provision on discharge, which was consistent with carer reports of a lack of support and communication by the health service [10]. However, Ramazanu and Griffiths found that carers of elderly hip fracture patients were stressed resulting from pressure to abide by routines to aid the injured persons recovery [11]. Whilst O’Callaghan [10] found that uncertain prognosis inflicted anxiety for carers; denial about the future was protective for some TBI carers who were consequently optimistic.

Research investigating the financial and employment impact of caregiving, particularly for people with musculoskeletal conditions, or polytrauma, is limited and has largely focused on US veterans [12,13]. Griffin, Van Houtven et al. found that carers with fewer financial resources experience greater carer burden, whilst those who are older and on lower wages are more likely to leave the labour force [12,13].

In a cross-sectional survey of 564 carers of TBI and polytrauma veterans, Griffin et al. [12] found that greater social resources mitigated the caregiver intensity-caregiver burden relationship. This shows the impact that social support can have on caregiver’s experience. However, the use of structural equation modelling and cross-sectional survey data makes it difficult to be certain of the direction of this association. It is possible that caregiver burden negatively influenced perceptions of social resources amongst carers, such that those experiencing a higher carer burden perceived there to be less support available.

Negative social interactions and a lack of social integration have been found to be significant independent predictors of depression in carers of a person with spinal cord injury (SCI) [8]. Despite this, research exploring the social impact of caregiving in other injury types is lacking (e.g., for orthopaedic injuries). The challenges and demands of caring for older people are well documented; however, a recent review of systematic reviews by Spiers et al. [14] stated that current evidence fails to quantify the impact of caregiving on wellbeing. Authors suggested the need for a comprehensive review to explore the impact of caring for older people on outcomes in terms of social and financial well-being. A different review by Bom et al. [15] explored the impact of caring for older adults, but compared findings across different subgroups, such as gender, marital status, type of care provided and other characteristics. The review concluded that married, female carers had poorer health outcomes than male carers, and increased intensity of caregiving generally led to worse health outcomes. However, it is still not clear why certain subgroups are impacted more than others.

Research investigating the financial and employment impact of caring for people with orthopaedic injuries and SCI highlights the extent of the economic burden. Findings show that carers are required to leave work, take extensive time off, or work flexibly, in order to retain their jobs [15,16]. In a survey of 99 orthopaedic trauma carers in the USA [17], caring responsibilities were found to affect capacity to work in 53% of carers, whilst 8% quit their jobs. Research typically investigates shorter term impacts following injury, despite evidence of financial burden continuing over many years [18]. It is well known that unpaid caring for older people can lead to a loss of employment or reduction in working hours [14], with many carers having to manage employment alongside caring responsibilities. However, there is limited evidence to describe the impact of this dual role on financial and social wellbeing not only in the widely studied older person carer population [14], but also in the traumatic injury carer population.

Theoretical models, such as The Roy Adaptation Model (RAM) [19,20] and the International Classification of Functioning, Disability and Health (ICF) [21] can be used to explore and describe the effects of caregiving and its impact on all aspects of life. The RAM describes humans as adaptive systems (physical, emotional, social, and functional) who respond to environmental stimuli through coping processes (adaptive modes). The model proposes that responding to stimuli via these adaptive modes influences a person’s ‘adaptation’ or adjustment to being a carer. Whilst the RAM focuses on personal psychological, and social factors, the ICF is a framework for describing and organising information on functioning and disability, recognising the impact environmental factors have on an individual’s life (Figure 1). The contextual factors (personal and environmental) can aid understanding of the impact taking on a caring role has on everyday life. Used alongside the RAM, the ICF allows for the exploration of a range different consequences of both the injury and carer giver roles (e.g., loss of job), including biopsychosocial, environmental and personal factors.

Conducting qualitative research can help to understand the impact of caregiving for the wider trauma population in more depth, compared to a quantitative survey. Several studies have explored the burden on caregivers of patients with TBI, cancer, posttraumatic stress disorder and older people; however, few qualitative studies have focused on orthopaedic trauma. Newcomb and Hymes [22] also identified this gap in research and interviewed 12 carers of severely injured orthopedic trauma patients. The purpose of this study was to identify stressors and burdens experienced by these caregivers, which included turmoil of life in hospital, difficulty obtaining and understanding information, and a sense that family was not considered to be an important part of the patient’s care plan. Although authors made a valuable contribution to the evidence base, they did not fully consider the impact on financial situation and employment, noting one limitation as recruiting all carers from a wealthy area of the country.

There is a clear lack of qualitative studies involving carers outside the two most prevalently researched injury types (SCI and TBI), and research investigating employment and financial impacts across injury types. Caregivers of trauma survivors are likely to be younger, income-producing individuals compared to often older carers of those with other conditions such Alzheimer’s disease, cancer, stroke. Therefore, there is a timely need to understand the perspective of carers, particularly given the uniqueness of each injury and its consequences. Therefore, to address the research gap, this study aimed to explore the impact of being a traumatic injury carer on finances, employment, social life and psychological wellbeing.

## 2. Materials and Methods

In this qualitative study, semi-structured telephone interviews were conducted with informal carers to understand participants’ experiences of caring for a traumatic injury survivor. Ethical Approval to conduct the interviews was obtained from the University of Nottingham Faculty of Medicine and Health Sciences.

### 2.1. Recruitment

Participants were recruited via traumatic injury survivors previously involved in the ROWTATE study (NIHR funded Ref: RP-PG-0617-20001, www.ROWTATE.org.uk (accessed on 14 September 2022) [23] that had consented to be contacted about future research. Previous study participants were contacted by author J.K. who asked if they were happy for their current or previous carer to be invited to participate in an interview. If a carer was nominated, the researcher (C.H.) emailed carers, providing a participant information sheet explaining what the study involved. Those interested in taking part were asked to respond to the researcher via phone or email. A recruitment poster was also posted on social media (i.e., Twitter), providing researcher contact details and asking carers to get in touch if they were interested in taking part.

Participants were eligible to take part in the study if they: (1) were (or had been) an unpaid informal carer for someone with a moderate to severe traumatic injury (defined as someone who spent at least two days in hospital following injury); (2) were aged over 18 years; (3) were sufficiently proficient in English to be able to take part in an interview; (4) were willing to provide informed consent.

Prior to recruitment, the researcher checked that potential participants had read the information sheet and had the opportunity to ask any questions. If individuals were happy to proceed and met the inclusion criteria, verbal informed consent was obtained from all participants. A consent form was completed by the researcher (C.H.) on behalf of the participant and a copy was sent to the participant.

We aimed to recruit between 6 and 10 participants, in line with Braun and Clarke’s [24] recommendations for small study interviews. Recruitment continued until theoretical sufficiency was reached.

### 2.2. Data Collection

Following consent, participants were asked to complete a short demographic questionnaire (Appendix A). Information collected included age and gender, pre- and post-injury employment status, injury of the person they care/cared for.

The interviews were conducted via telephone. The semi-structured line of questioning allowed participants some flexibility in discussing areas of preference related to the topic of being a carer. The topic guide was informed by two theoretical frameworks: the RAM [19,20] and the ICF [21]. This gave the line of questioning a scientific basis (see below for rationale).

Topics discussed during the interviews included: (1) the impact of caring for a trauma survivor on the carer’s employment; (2) financial impact of the injury to the household, and the interaction of these things with the carer’s psychological wellbeing; (3) the impact of caring on social life was also discussed in relation to psychological wellbeing. The topic guide is shown in Appendix A.

Interviews were audio recorded and transcribed verbatim. Any identifiable participant information was anonymised.

### 2.3. Analysis

Interview data were thematically analysed [25] using NVivo version 12, informed by the ICF and RAM. A summary of the analytical process is shown in Table 1.

Using two theoretical frameworks that complemented each other, ensured all aspects of the research question were covered. Whilst the RAM focuses more on personal psychological, and social factors associated with caring, the ICF enabled investigation of the impacts of caring on employment, financial situation, and access to services, through its focus on environmental factors, such as workplace support. Although the ICF is commonly used to describe the functioning of an individual with a health condition, here we have used the framework to consider contextual factors linked to being a carer and the impact this has on their life. This not only enabled a consideration of personal factors that contribute to psychological wellbeing, but also specific environmental factors that hinder or improve the carers life. The coding framework is shown in Table 2.

Once key themes had been identified by author CH, findings were discussed with author J.K., for agreement. Analysis aimed to identify links between findings related to each of the ICF contextual factors and adaptive modes of the RAM.

## 3. Results

Ten carers of people who sustained a traumatic injury were interviewed (n = 9 females, n = 1 male). Interviews lasted between 50 and 85 min. The mean age of carers was 52.4 years (SD 12.9). The injuries of the people participants care for occurred between 6 months and 10 years ago (mean 4.44 years ago, SD = 3.1). At the time of recruitment, two participants were in full-time employment, four were employed part-time (one on long-term leave), one was retired and three were unemployed. Prior to taking on their caring role, two of the unemployed participants were in full-time employment, and one had to reduce their working hours from full-time to part-time. Participants were employed for an average of 21.75 (SD = 17.25) hours per week prior to their caring role, which reduced to an average of 12.5 (SD = 15.29) hours per week after becoming a carer. Five participants had to reduce their working hours as a result of their new role. A summary of carer characteristics is shown in Table 3. Participants cared for individuals with a range of injuries. Five injured persons (50%) had suffered polytrauma, one of whom also sustained a TBI, and one sustained a spinal injury. Two individuals suffered a TBI, one a fractured ankle, one a fractured elbow and one upper limb injuries. Three individuals had sustained their injury in the past 6–12 months, five over 4 years ago and two over 10 years ago. The majority of participants (n = 6) spent the whole week (i.e., 7 days) caring for the injured person and some (n = 2) were in their caring role for only two days per week.

Interviews identified three key themes: (1) financial impact and employment issues, (2) relationships and support and (3) psychological impact. A summary of themes and subthemes are shown in Table 4.

### 3.1. Theme 1: Impact on Finances and Employment

#### 3.1.1. Employment

Most carers employed at the time of the injury felt that their employer was understanding and spoke about the positive support they received:


*“I was off for months whilst [trauma survivor] was in hospital. I think I was off for the first four months…work were very good. They just said, go on long term sick and just get your GP to keep signing you off. They paid me so I didn’t have any stress related to employment. And that’s probably a joy of being in the public sector.”*
(P8, carer for ankle fracture)

In the immediate aftermath of the injury, employers were keen to keep carers in their job, despite many taking a significant amount of time off work. Receiving sick pay in addition to time off relieved financial pressures and reduced the potential for negative consequences resulting from financial worries. Carers felt that their caring role took priority and appreciated the understanding of their employer. Theoretically, the response of employers (ICF environment) impacted on carers’ Functional mode (RAM), as they put less effort into work and concentrated efforts on caring, thus reducing caregiver strain. Generally, carers’ jobs were secure. However, most were employed in the public sector or were qualified professionals (see Table 2 for further detail).

For one self-employed individual, their new caring role posed a significant challenge:


*“It’s tiring cause you know, you’re rushing down there [work] to sort out things, but then know that I need to be back here for something… sometimes I knew I wanted to be at home, if [trauma survivor] didn’t have a very good day or something. And I wanted to be here [home], but I’d need to be there [work].”*
(P4, carer for polytrauma)

Taking time off work was not always an option for some carers. Owning a company made certain aspects of work impossible to delegate to employees.

#### 3.1.2. Costs of Rehabilitation

Most carers experienced significant financial burden. The greatest costs were incurred by home adjustments and equipment for the trauma survivor:


*“We have a downstairs playroom that was fitted with a hospital bed. We then had grab rails fitted in the downstairs toilet…we had an extra handrail fitted on the stairs going upstairs and we decided what we would do is to completely redo our bathroom …We basically spent twelve grand redoing the bathroom.”*
(P6, carer for TBI)

Socioeconomic differences present among carers affected the acceptability of these costs. Whilst all viewed such expenses as an undisputable requirement to support recovery, some families felt less financial impact. For such carers, spending money on the trauma survivor’s rehabilitation was a proactive, straightforward way of supporting recovery. However, for others the costs presented an unanticipated financial burden:


*“You have to have an access point to get [trauma survivor] into the house with the wheelchair. You’ve gotta have a bed, such a height—so he can get on and off it…I had patio doors at the front of the house, got a ramp, I think it costs me £2100.00,”*
(P4, carer for polytrauma)

Certain equipment had to be installed prior to hospital discharge, which resulted in financial concerns and raised anxiety for some carers. Specification regarding equipment requirements also removed choice from carers about how much money to spend. Instead, they felt compelled to abide to recommendations even if this resulted in financial debt.

#### 3.1.3. Difficulty Receiving Financial Support

All carers struggled to navigate the receipt of state benefits associated with the injury’s occurrence:


*“Some of the most stressful things we’ve been through in recent years have been problems with benefits. And you’ve got you’ve got nobody to turn to.”*
(P4, carer for polytrauma)

Accessing financial support from the state was complex with no help available. The responsibility landed on carers, exacerbating burden. For some this increased their sense of isolation due to the healthcare system’s lack of involvement, which led carers to believe they were not entitled to receive financial support:


*“I’ve no idea where we go with how benefits work in that situation...I don’t know who we would turn to…We were very lucky that in the end, you know, for the first five years we had people we could turn to”*
(P7, carer for TBI)

The absence of any clear guidelines about how benefits fitted with recovery progression meant that carers were concerned about the trauma survivor’s future. This was particularly the case for TBI carers who received support from a social worker for the first five years post-injury which then stopped.

#### 3.1.4. Attitude towards Work

There was a contrast in attitude towards work in younger and older carers. For older carers who were often in less demanding roles, work was deprioritised. They were able to relax efforts, but also expressed a lack of interest in concentrating on work during the trauma survivor’s recovery period.

Instead, their priority was being present to provide emotional and practical support:


*“Do you know, I just did them. I did what I had to do. But I knew that the people at the livery yard were hands on, so I felt confident I didn’t feel, I didn’t feel it was a worry at all. I was just concentrating on [name of trauma survivor]”*
(P3, carer for polytrauma with spinal injury)

However, younger carers often felt conflicted:


*“Yeah, I mean it was definitely draining. And obviously I felt very split because it’s worked out one of those things…like it wasn’t difficult for me to just drop it all because my main priority was my Mum. But I’m also very hard working and I want it to do well at work,”*
(P2, carer for polytrauma)

Although supporting the trauma survivor was prioritised by all carers, younger individuals were more eager to stay at work. There was a sense that the injury posed a barrier to career success and progression. Rather than being pushed aside, for many young carers, work was something they attempted to juggle simultaneously with caring responsibilities, often resulting in exhaustion.

### 3.2. Theme 2: Relationships and Support

#### 3.2.1. Social Life

Carers talked about a close bond between themselves and the trauma survivor, however the same was not reflected in other relationships, such as friends and family. Often such relationships suffered due to their caring role. Several carers expressed difficulty trusting others following the injury:


*“I’m getting there, I can walk into a room with strangers now and be a little bit “oooh”—you know, who are these people? Are they nice…a first instinct sort of anxiety.”*
(P1, carer for polytrauma)

Anxiety about leaving the house and socialising with others appeared common among carers. Some participants became apprehensive about re-engaging with friends. Despite this, many carers experienced support from friends, even if peripheral:


*“Largely people want to help, they want to be involved. I don’t know if it’s a certain type of person that we’re friends with… but everyone is very supportive…I imagine it depends on how close you are, it affected our daily lives completely, probably the lives of other people less so.”*
(P8, carer for ankle fracture)

Carers valued the offers of help from friends. Such friendships were helpful both practically and emotionally, especially in the early stages post-injury.

However, despite some carers letting friends help post-injury, for others this was too difficult in the immediate aftermath:


*“It kind of just stopped straight away and I found it hard to—because obviously the first time you see someone, it’s like the whole having to talk and address it.”*
(P2, carer for polytrauma)

Many experienced social awkwardness resulting as they did not want to constantly relive the trauma. This made it challenging to maintain relationships with friends, increasing carers’ sense of isolation.

#### 3.2.2. Family Dynamics

Many carers talked about the value of close family and the additional support they offered:


*“I did have the discussion with our son before my partner had come home. And I said, look, if your dad needs physical care, what’s your thoughts on that? And he actually said, I’ll do it. So, it was a backup that, you know…he was willing and happy to take on that role.”*
(P10, carer for upper limb injuries)

The offers of support from family relieved pressure for carers. However, none assumed that someone else would be primarily responsible for providing care. This relates to the knowledge they gained about how to support rehabilitation, demonstrating a relationship between the RAM Emotional/Self-concept mode and ICF Personal factors, as carers’ superior insight affected their willingness to accept help.

In some cases, offers were declined by carers. Instead, many were more likely to accept offers of practical help from family members unrelated to caring (e.g., collecting children from school):


*“Obviously, I’ve got a young son. So, he basically dipped in and did everything that needed to be done…So actually it was a pressure off, I haven’t got all of that to worry about as well. He was there, you know, picking up from school and sorting all of all of that out. So, yes, a big indirect help.”*
(P9, carer for elbow fracture)

Such help facilitated carers’ ability to support the trauma survivor by reducing other stressors associated with family life. Carers struggled to juggle different family relationships because of their commitment to the trauma survivor.

However, others talked about the negative experiences they had, with some expressing frustration that the caring responsibility fell solely on them:


*“It’s the impact that the relationships between you and other family members, that can cause a lot of—can add to the trauma. It puts a real strain… I was kind of the only one that would go in and see [trauma survivor] to start with. And that can put a lot of strain on. You know, I was the one that kind of took the lead in terms of trying to speak to consultants”*
(P2, carer for polytrauma)

Some carers appeared to fall into the role of being the primary care provider. As a result, tension rose within families. Despite a clear commitment to supporting trauma survivors, for some, there was a sense of being let down by family. Many carers felt overwhelmed by the expectations placed upon them.

#### 3.2.3. Psychological Support

Carers had different experiences of accessing psychological support. This was often dependant on the injury type:


*“One of the things that [wife] did that I didn’t, and regret, was that the neuropsychologist at [hospital], said you know I am available to support the family and it’s you know, that’s part of the MDT [multidisciplinary team] whole family approach.”*
(P6, carer for TBI)

In the case of TBI, psychological services were offered to family members. This support gave carers the opportunity to discuss the impact of injury on them and consequently move forward with caring responsibilities. However, this support did not extend to other injury types (such as orthopaedic injuries) which resulted in some carers feeling neglected and left to cope with the trauma of the injury alone:


*“That is the bottom line, there is not the profession support there, even one appointment, even to be offered one appointment after such a traumatic event, a few weeks down the line, you could get it out of your system. And they could gauge what the repercussions of the accident were on your mental health.”*
(P5, carer for polytrauma with TBI)

#### 3.2.4. Healthcare Support

Carers that worked as healthcare professionals pre-injury were often more involved in discussions regarding the trauma survivor’s care. This meant they had a greater insight into the impact of injury and received more information about rehabilitation services. This appeared to contribute to the level of support carers received, with a greater awareness of services that could be accessed in the longer term:


*“And I think, you know, they were very helpful. I don’t know whether some of that was also due to the fact of being a healthcare professional...the doctors and people, they were perhaps a bit more ready to discuss things with another healthcare professional than they would have been with somebody who maybe struggled to take in what they were saying.”*
(P7, carer for TBI)

Carers often felt abandoned post-discharge and left to cope with the daily challenges of providing care*:*


*“…before [trauma survivor] left hospital, no one had told me, he might be like this, or he might be like that…The other thing was you must keep him hydrated, which was easier said than done. When you’re putting glasses of water, tea by him—I don’t want to drink those. What do you do? Pour them down his throat and choke? On the one hand you’re being told one thing, but not how to execute it.”*
(P5, carer for polytrauma with TBI)

### 3.3. Theme 3: Psychological Impact of Caring

#### 3.3.1. Acceptance

Most carers struggled to come to terms with the injury:


*“I think in the very early stages, it’s very hard. You know, when they give you that literature from Headway [brain injury charity] and things and you don’t really you don’t really want to accept what the possible future is.”*
(P7, carer for TBI)

This difficulty was exacerbated for TBI carers where the prognosis often included lifelong disability. Carers’ initial distress related to the impact of the injury for the trauma survivor excluding the effect it would have on their lives. Other carers were able to come to terms with the consequences more quickly:


*“That situation is something the patient and the carer or next of kin can’t get out of, but you’re stuck in that situation, that’s your life… You can’t shut the door and just visit it once in a while like everybody else.”*
(P1, carer for polytrauma)

#### 3.3.2. Emotional Impact

Adjusting to new routines was psychologically difficult for some carers. Carers felt a weight of new responsibility and pressure, often concerned that they might hinder the trauma survivor’s recovery if they did not support them.

Despite the challenges, carers expressed overwhelming emotion and happiness when rehabilitation goals were met:


*“I remember being in the office when I got a phone call from [name of wife] to say the physios rung, [trauma survivor] just walked across the length of the gym, in an [argent] walker bursting into tears in the arms of my boss. This is not what we normally do in the office.”*
(P6, carer for TBI)

### 3.4. Summary

A visual summary of findings is given in Figure 2, showing positive and negative responses from carers. This provides an overview of all data collected, however only the key themes have been presented in text. This figure highlights the relationship between the two theoretical models and illustrates how contextual factors (i.e., environmental and personal factors of the ICF) are linked to the adaptation modes of the RAM. The negative impacts of caring and therefore barriers to adaptation are shown in red in Figure 2, including financial costs of rehabilitation, lack of communication with healthcare professionals and family tension. These were identified as things that affected a carers ability to adapt to their new role, which in turn negatively impacted on their wellbeing. Linking to the contextual factors of the ICF, barriers to adaptation were worsened by environmental factors such as inconsistent access to psychological support, issues with social services and difficulties accessing financial support. However, a supportive employer was identified as something that helped carers to deal with their new role and in many cases, meant they could continue their employment. Several positive impacts and facilitators to adapting to the caring role were also identified, including support from family, building a good rapport from healthcare professionals, and having a close relationship with the trauma survivor. All of these helped the carer to adapt to their role. In terms of personal factors that contributed to positive impacts of caring were having a good understanding of how to support injury recovery and being aware of limitations.

## 4. Discussion

This study provides some insight into the impact of being a caregiver for a person with a traumatic injury, on family finances and carer’s employment, and social life and psychological wellbeing. Findings suggest that individuals step into the role of primary care provider and for many injured people, caregivers were not supported by healthcare services post-discharge. They also felt solely responsible for looking after the trauma survivor despite offers of support from friends and family. This counters evidence which suggests greater social resources mitigate the caregiving intensity-caregiver burden relationship [12].

Several findings related to support were noted which have important implications, such as limited psychological support for carers across injury types, and inconsistent care and communication in hospitals. Support was especially limited for carers supporting a person without a TBI, particularly regarding the provision of rehabilitative care; whereas in the case of TBI, professional care in the community was given, this corroborates existing evidence [26].

Carers viewed offers of help from friends and family as temporary and peripheral to the trauma survivors’ recovery. This stemmed from challenges navigating the hospital environment and carers finding it easiest to take sole responsibility for interactions with hospital staff. Reluctance to share caring responsibilities is consistent with others’ findings, indicating poor involvement of families in post-injury care plans [27]. In this qualitative study, carers found the hospital environment confusing and challenging to obtain information in, whilst the family were not considered important in care planning. The findings of our research expand on this as carers consequently experienced close relationships with trauma survivors whilst others suffered, resulting in family tensions.

Carers who were also healthcare professionals were given more information about the injured person’s recovery trajectory and were able to build rapport with doctors. Thus, enabling them to access more long-term support. This contrasted other carers who received minimal or unprofessional communication from hospital staff [28,29]. Furthermore, whilst research indicates that TBI carers look to the future once the trauma survivor has returned home, and experience anxiety at a lack of information provision [28]; our findings captured a mixed response with some TBI carers initially struggling to come to terms with the injury. This is in line with research indicating changes in information needs during the transfer from primary to secondary care [29]. Such changes along with the unequal treatment of carers who are healthcare professionals highlight the need for standard information provision across different recovery stages, and consistency in carer communication. Interventions should seek to increase carers’ sense of self efficacy to forge relationships with hospital staff to access greater support.

The impact of the injury on carer employment as well as financial consequences is consistent with the literature [13,16]. Employers responded supportively, meaning that carers were able to keep their jobs, and prioritise caring responsibilities; although differences were present for those not employed in the public sector. Whilst older carers prioritised work less, this was partly related to their employment status (e.g., part time) rather than solely the employer’s response. However, they did not all leave the workforce [13].

Although literature highlights the link between financial concerns and depression among carers [30], in this study, such impact appeared more of a concern for less financially stable families (i.e., lower socioeconomic status). In line with studies which have identified financial stress experienced by carers [31], this was associated with difficulties receiving benefits. Difficulty accessing financial support was universal across carers. A lack of help in completing applications for state benefits and compensation increased carers’ sense of isolation and burden. This corroborates research which identifies a need for compensation advice at major trauma centres [32]. Caregivers of individuals with more complex injuries and those with TBI appeared to be affected more financially than those caring for individuals with less complex injuries and shorter recovery periods. For individuals with longer recovery trajectories, the greater the financial impact on caregivers as they were forced to reconsider their employment and ability to stay in work. The healthcare service should provide information and assistance to families regarding financial support following injuries. This may reduce unnecessary burden from carers and match the support that was offered by employers.

Many carers struggled to accept the injury, particularly when this was associated with severe physical and in some cases mental disability [2,3]. This contributed to negative emotions. However, in some cases denial about the long-term prognosis helped carers to be upbeat and motivated regarding rehabilitation. Findings suggested that carers of those with less severe injuries, such as single limb fractures and shorter recovery trajectories were less psychologically affected than those who cared for individuals with more severe and/or long-term injuries, such as complex polytrauma and TBI. Perhaps negative psychological impact was linked to not knowing when the person they care for would recover, or if they ever will recover, and a lack of education around the impact of injury. Our findings align with research involving informal carers of stroke survivors [33], suggesting that psychosocial and educational support (e.g., ensuring the caregiver understands individual needs) should be provided for stroke caregivers to reduce daily caregiver burden and improve psychological wellbeing. Considering our findings, healthcare workers should seek to communicate the recovery trajectory of injuries gradually to help carers process and respond to the injury in the present tense, along with educating them about impairments and potential long-term needs. This would improve information provision regarding prognosis, which is currently lacking [34]. Future research should seek to investigate the relationship between carer attitude and the impact on emotions at different stages of the recovery trajectory.

This study had several strengths. The broad sample enabled some comparison to be drawn between well researched conditions such as carers of people with TBI and under researched groups. Many of those interviewed were carers of people who had sustained multiple injuries (polytrauma) where no head injury had been sustained. We recruited a sample of carers whose lives had changed dramatically, with 50% having to reduce their working hours or relinquish work altogether. This addresses a gap in research regarding the impact of caregiving for the trauma population on work. Additionally, using two theoretical frameworks enhanced the scientific rigour of analysis and allowed different areas of the research question to be addressed in relation to theory. However, there were some limitations. The sample was small and almost all female, which limits the generalisability of conclusions. It also limits the ability to make strong comparisons between caregivers of different injury types. Our sample size included a number of older women, some of whom were retired, which again limits findings. Future research should therefore attempt to recruit more male carers and carers of working age.

Due to the COVID-19 pandemic, we had to conduct interviews remotely via telephone, which may have introduced recruitment bias. We understand that limiting the interview to telephone may have excluded carers that do not have a phone or may be less able to communicate remotely in comparison to a face-to-face interview. However, we feel we were still able to recruit a diverse sample of carers with a range of experience despite this. Another limitation was not using a specific definition of a carer (including tasks links to advanced activities of daily living, personal activities of daily living and intensity of care provided) as part of the inclusion criteria. This would have enabled us to provide a clearer description of the sample. We did not collect data on ethnicity and while the intention was not to focus on Caucasian carers, the methods of recruitment did not attract an ethnically diverse sample. This is an unrepresented sample and we recognise it as a limitation. Further research should adopt different recruitment methods to better understand the impact of caregiving from the perspective of people from a broader range of ethnicities and socioeconomic backgrounds.

Our interviews only provided a snapshot of the carers’ situation and experience at one point in time and while both theoretical models assume an interaction between the patient or carer and the context, they do not necessarily account for interactions with others or changes in the context over time [35]. Therefore, further research is needed to explore these interactions, along with the expectations for the future of these trauma survivors. Research is required to better understand the longer-term impact of caring for someone with serious injury, in terms of the in length, intensity, and type of care, but also in terms of the socioeconomic impact on work, income, family finances and the wider economy. Given that the majority of admissions to UK major trauma centres are people aged over 65 and the known association between care giving and negative health effects particularly in older women, research should explore the longer-term impact of caregiving following serious injury and its relationship with ageing in carers, thus building on existing evidence [15].

## 5. Conclusions

This study highlights the significant life changes experienced by carers of trauma survivors with a range of injuries. Support for carers outside the hospital setting is limited for people who have not sustained a TBI. Employers were supportive of carers, which reduced financial stress and caregiver burden. Considering the findings through a theoretical lens (via the RAM and ICF) enabled the relationship between contextual factors and adjustment to be explored. To the authors’ knowledge, combining these theoretical frameworks is a novel approach to presenting carer research in this field and should be applied when developing interventions, thus ensuring the full impact of caregiving is considered. In light of the findings, funding should be spread across injury types for post-hospital care and carers should receive support navigating state benefits and accessing funding. Those commissioning or managing healthcare services should acknowledge the psychological impact of caregiving and provide further support, particularly to manage carer burden. Future studies and interventions should aim to standardise communication with carers regarding injury recovery and rehabilitation service provision, improving carer education and awareness. 

## Figures and Tables

**Figure 1 ijerph-19-16202-f001:**
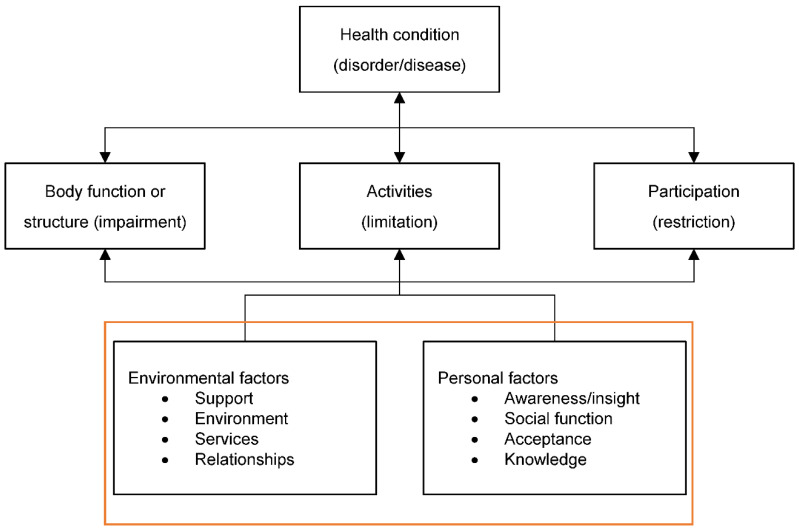
International Classification of Functioning, Disability and Health (ICF) with expanded Environmental and Personal Factors (used in the coding of the interview transcripts).

**Figure 2 ijerph-19-16202-f002:**
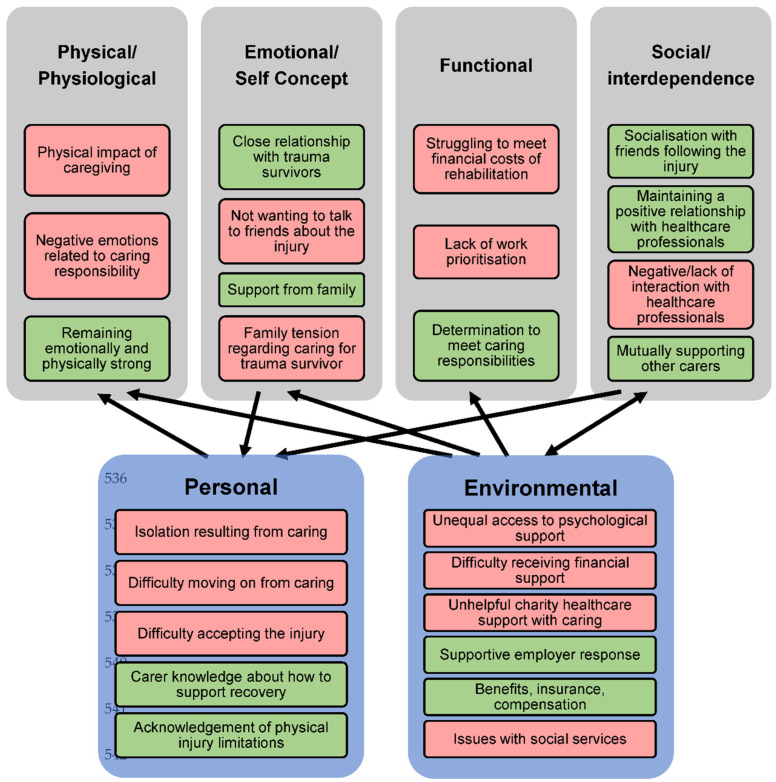
Theoretical frameworks of the RAM and ICF Contextual factors with positive and negative findings and relationships between the framework components shown. Red boxes indicate a barrier to adaptation or negative effect as a result of caring, green boxes indicate a facilitator to adaptation or positive effect as a result of caring. The arrows indicate relationships between the RAM and ICF models, showing that different factors can influence one another.

**Table 1 ijerph-19-16202-t001:** Description of theoretical thematic analysis process.

Phase	Description
1. Familiarising yourself with your data	The data were read through twice initially and key similarities and differences across participants experiences were noted.
2. Applying theoretical frameworks to the data	The full data set was uploaded to NVivo 12 software and themes and subthemes from the RAM and ICF frameworks entered. Quotes were highlighted according to these showing the relevance of the frameworks to the interview topics. Additional themes were also identified as present in the data and entered into the software alongside highlighted quotes.
3. Breaking down frameworks and understanding the data fitted	Quotes attached to the RAM themes and subthemes of the ICF factors as well as additional new themes were grouped on word documents and printed. Subthemes were then identified from within the RAM and specific groupings of quotes from the ICF subthemes were highlighted. This aided a more in depth understanding of the data.
4. Interpreting the findings and generating themes and subthemes	Key quotes from across the data were extracted and interpreted resulting in the creation of new themes and subthemes which reflected the findings.
5. Refitting theoretical models to the data and creating a diagram	Themes from the RAM and ICF were reapplied to the data and a diagram was created containing positive and negative aspects of the findings. This included the new subthemes and general findings from the data. Links were identified between the theoretical themes. This aided discussion of the findings.
6. Creating a table of themes and subthemes	From the findings a table of themes, sub-themes and short summaries of each theme was created capturing meanings. This included subthemes the reporting of which was beyond the scope of the paper.

**Table 2 ijerph-19-16202-t002:** Summary of interview coding framework.

	Theme/Code	
RAM	Physical/physiological mode (how a person responds physically to environmental changes)	
	Self-concept/emotional mode (person’s beliefs and feelings about himself/herself)	
	Functional mode (behaviours associated with one’s position role in society)	
	Interdependent/social mode (one’s relationships and interactions with others, feelings of love/value)	
ICF CONTEXTUIAL FACTORS	Environmental factors	Support (any support from healthcare professionals, family members, support workers, charities other)
		Environment (surrounding environment and the impact this has on ability to care for patient, ability to work, mental and physical health)
		Services (any services that are available to support a caring role or ability to work—healthcare, social, job centre etc.)
		Relationships (relationships between carer and patient, therapist, employer and other family members)
	Personal factors	Awareness/insight (insight and awareness of impact of patient’s injury, aware of recovery trajectory and impact on carer’s life)
		Social function (interactions with environment, engage with work, social activities, relationships)
		Acceptance (Acknowledging or accepting changes in life as a result of caring, moving forwards)
		Knowledge (knowledge of carer, employer and family members about patient’s injury and how to support them, knowledge of available support services)

**Table 3 ijerph-19-16202-t003:** Summary of carer participant demographic information. TBI: traumatic brain injury, NA: not applicable.

Unique ID	Age (Years)	Gender	Number of Days Spent Caring per week	Relationship to Person They Care for	Current Employment Status (i.e. Whilst in Caring Role)	Pre-Caring Profession/ Role	Employment Status Changed since Caring Role?	Pre-Caring Employment hours per week	Current Employment hours per week	Injury of Person Caring for	Time since Injury of Person Carnig for
1	50	Female	7	Wife	Unemployed	NA	No	NA	NA	Polytrauma (orthopaedic and nerve injury)	Over 4 years
2	32	Female	5–7	Daughter	Full time	Pension’s consultant	No	35+	35+	Polytrauma (severe pelvic injury, punctured lung, leg and shoulder fracture)	Over 4 years
3	68	Female	7	Mother	Retired	Farm owner	No	NA	NA	Polytrauma (leg, spine, pelvis and facial injury)	6–12 months
4	60	Female	7	Wife	Unemployed	Bakery owner	Yes	40+	0	Polytrauma (several major fractures)	Over 4 years
5	67	Female	7	Wife	Part-time	Outsourcing secretarial work	No	5–6	1–2	Polytrauma with TBI	Over 4 years
6	60	Male	2	Parent	Part-time	Compliance Officer	Yes	40	28	TBI	Over 10 years
7	61	Female	2	Parent	Part-time	Health Care Professional	No	22.5	22.5	TBI	Over 10 years
8	31	Female	Not known	Partner	Full-time	Lecturer	No	37	37	Ankle fracture	6–12 months
9	41	Female	7	Daughter	Unemployed	Civil Servant	Yes	37	0	Open elbow fracture	Over 4 years
10	54	Female	7	Partner	Part-time (on long-term leave)	Health Care Professional	No	0	0	Upper limb injuries	6–12 months

**Table 4 ijerph-19-16202-t004:** Summary of interview themes.

Main Theme	Sub Themes	Theme Description
Relationships and support	Social life Family dynamics Psychological support Healthcare support	Carers were committed to aiding recovery and possessed close emotional bonds with trauma survivors. This took a physical toll on them although involvement in care meant that they were constantly in tune with recovery progress. Some carers received emotional and practical help from close friends however others struggled to discuss the injury. Carers sought comfort in those who had shared their experience whilst in hospital. Families were mostly supportive although tensions related to the carers role. Psychological support was only offered in the case of TBI and carers who were also healthcare professionals received preferential communication and involvement whilst in hospital settings. There was a lack of information provision regarding how to carry out care at home. Communities responded favourably finding ways to offer support beyond practical caring.
Financial and Employment impacts	Employment Costs of rehabilitation Difficulty receiving financial support Attitude towards work	Most employers responded generously to the injury enabling carers to focus on the injured person. Rehabilitation was associated with various costs including hospital car parking and home adjustments which were more acceptable to higher socioeconomic status carers. Receiving financial support was difficult to navigate and unsupported although several carers received finance from external sources. Younger carers were more motivated to prioritise their jobs which conflicted with caring.
Psychological impact of caring	Acceptance Emotional impact	Carers varied in terms of the ease with which they accepted the injury’s occurrence and the impact it would have on the family. Self-pressure to deliver care resulted in negative emotions and some carers even struggled to adjust to the trauma survivor’s recovery. Frustration was expressed about the loss of hobbies although most carers accepted a temporary hiatus to their social lives. Carers also acknowledged ways in which they experienced privilege compared to others in their situation.

## Data Availability

The data that participants have consented to share are available to potential researchers. Requests detailing the research aims and use of the data should be sent to the corresponding author via email: jade.kettlewell2@nottingham.ac.uk.

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
