# Peer review of "A Qualitative Study to Understand the Impact of Caring for Traumatic Injury Survivors"

_ijerph, 2022, doi:10.3390/ijerph192316202_

Round 1

Reviewer 1 Report

The article is a study of impact caring for traumatic injury survivors has on their carers. The study is well designed and articles is well structured. However some parts of the article would need to be more developed and some inconsistencies in presentation of results would need to be addressed  before publication.

Introduction present the caring burden, however seems limited to only few studies.  This would be good to be further strengthened with additional sources. Perhaps some similarities could also be drawn from caring burden of employed carers  well researched in caring for older people – and its effect on employment, psychological and social effects, than specifying specifics of the caring burden of the trauma survivors.

It would be welcome to also more specifically define the carer – usually it is defined by the tasks they do (AADL – advanced activities of daily living,  PADL, personal activities of daily living) and intensity of care provided. Also  a little more description on diversity of these carers – presumably they are adults and not young carers (as that would entail specific issues related to care burden), and if there is a general profile of these carers or not.

It would be good to provide a more in depth description of the theoretical models applied (Roy adaptation model and its adaptive modes in more detail, RAM is not described) and also in what part they will be used and how they relate to each other (how they complement each other). Also, it remains unclear how these models were used for the analysis – this should be better elaborated also in the description of theoretical analytical process (what themes and subthemes from the two theories were used exactly, ...)

Classification of functioning, disability and health theory was used, however it should be clearly argument its reference for carers and how its was used.

The authors argue well the importance of looking at uniqueness of injuries and their consequences, but more elaboration would benefit the reader to understand the differences (in length, intensity, type of care, etc.)

The data method has been described well and in detail.

The description of the interviewed sample is not most detailed and I would suggest to report the numbers of those in full time employment and those not (as averages blur too much the picture as it looks major change and reduced but in reality for majority there was no change). Similarly post caring employments would be better described more in detail, as this has been also one of the focuses of the study (average again is not very informative). In table 2 it is unclear the caring employment – is that meant employment post injury (during caring employment) – perhaps improve this label.

Results

Not clear why physical impact of caregiving is included in relationship theme?

There seems to be an overlap between relationships and support theme – it should be explained how these two themes differ (as in relationship description you speak of practical help from social networks, which implies support?) This is evident also in discussion of support, where psychological support describes only professionals, but family and friends are not described here, which seems strange as we read elsewhere they could be important for psychological support. The authors should reconsider how these themes interact and how this overlap can be addressed.

Figure 2 is introduced early but not explained in the text. Elaboration of this figure is needed.

Some subthemes are not explored – why this decision was made and on what was this selection for in-depth description based on is not clear and should be made clearer to the reader.

In discussion the authors stress as strength of the study its ability to compare between TBI and other  less researched groups. However this strength is not clearly visible in the analysis itself, as there is almost no comparison of experiences between different carers. Furthermore, the small sample limits this strength therefore the authors might strengthen this comparative perspective in description of results but also emphasise the limits the small sample size puts on this comparison.

A deeper reflection of the usefulness and perhaps shortcomings of the used theoretical models would also strengthen this part.

Author Response

Thank you for taking the time to review our manuscript and your comments. Please see the attachment for our responses. 

Reviewer 2 Report

Dear Authors! Thank you for your paper. Different aspects of carers lifes are realy important for understanding as many of them underestimate what is happening due to the situations of caring about someone.

My only question  and comment would be, what is the difference between the psychological aspects of caring for different types of traumas? Is there any rationalle to expect different psychological effects for those who care about traumatic injury comparing to those caring about stroke or any other person who needs informal carer?

Author Response

(The authors gave the same response as above.)

Reviewer 3 Report

Hudson and co-authors investigated the topic of "A qualitative study to understand the financial, social, and psychological impact of caring for traumatic injury survivors", by applying the qualitative method. They specifically focused on the hypothesis to explore the impact of caring on family and caregiver finances, employment, social life, and psychological well-being. Although this is an important topic and sounding the readers, it has some important downsides:

The title is sounding and interesting, but it doesn’t correctly describe the main topic of this paper.

The sample size is too small to get a real conclusion or determine that current data represent the total population.

How does the author correct for bias? Since it was a phone interview.

What are the concerns and expectations for the future of these patients or survivors?

What is the impact of aging on injury survivors? Since most of the participants were female and older.

What was the main concern of traumatic survivors? 

Author Response

Thank you for reviewing our manuscript and for your comments. 

Please see the attachment for our responses. 
